# Understanding the Resilience of Different Farming Strategies in Coping with Geo-Hazards: A Case Study in Chongqing, China

**DOI:** 10.3390/ijerph17041226

**Published:** 2020-02-14

**Authors:** Li Peng, Jing Tan, Wei Deng, Ying Liu

**Affiliations:** 1College of Geography and Resources, Sichuan Normal University, Chengdu 610101, China; pengli@imde.ac.cn (L.P.); dengwei@imde.ac.cn (W.D.); liuying@imde.ac.cn (Y.L.); 2Key Laboratory of Land Resources Evaluation and Monitoring in Southwest, Ministry of Education, Sichuan Normal University, Chengdu 610101, China; 3China Western Economic Research Center, Southwestern University of Finance and Economics, Chengdu 610074, China; 4Institute of Mountain Hazards and Environment, Chinese Academy of Sciences, Chengdu 610041, China

**Keywords:** geo-hazards, farming strategies, adaptability, Chongqing, China

## Abstract

Adjusting farming strategies are adaptive behaviors to cope with hazard risks. However, few studies have studied rural and remote mountain areas in China with little known about “farmers’ adaptation under the impact of geo-hazards”. Unlike traditional farmers’ behavioral adaptation studies, in this study, we focused on the resilience of farmers’ behavioral mechanisms to address local hazards such as geo-hazards. Our data were acquired through questionnaire responses (*N* = 516) in mountainous hazard-prone areas in Chongqing, China. The binary logit model and multinomial logit model were used to investigate the obstacles to different farming strategies and the determinants of adaptation strategy choice, focusing on the effects of disaster experience and social support on the adaptation strategy resilience. The results show that the most common adaptation strategy was adjusting crop varieties, and the largest adaptation obstacle was a lack of funds. Additionally, the age of the smallholder, farming acreage, agricultural income, social support, and disaster experience significantly increased the possibility of farmers adjusting their agricultural production. Of these, smallholder agricultural income, state disaster subsidy, the presence of disaster prevention construction, the smallholder’s property, and the presence of disaster-caused crop loss experience were the most important factors affecting a farmer’s adaptation strategy. In particular, farmers were more sensitive to disaster-caused property loss than to disaster-caused crop loss. This study can provide implications for the government to formulate disaster mitigation measures and for farming strategies at the smallholder level.

## 1. Introduction

Landslides and mudslides are common geo-hazards in mountainous areas. China is a country with frequent geo-hazards, and the survey results showed that there are more than 290,000 sites at potential risk of geo-hazards in China; most are distributed in the mountainous areas of western China. Due to natural factors and human activities, geo-hazards have been increasing in recent years, causing increasingly serious impact on the production and lives of local people [1,2]. According to China’s ministry of natural resources, in 2018, 2966 geological disasters occurred in China, causing a direct economic loss of 1.47 billion yuan. Of these, 1044 incidents occurred in the southwestern region, accounting for 35.2% of the total incidents, with a direct economic loss of 660 million yuan or 44.9% of the total economic loss. Chongqing is the most well-known “mountain city” in southwestern China, which covers an area of 82.4 thousand square kilometers and 94% landscape in this area consists of mountains (75.8 %) or hills (18.2 %). Chongqing also has one of the highest numbers of hidden geo-hazards (16,412 potential sites identified) in China. In addition to causing casualties and loss in property and infrastructure, geo-hazards have resulted in soil degradation and reduced agricultural productivity, thereby hindering economic development in mountainous areas [3]. This phenomenon has had a more profound negative impact on farmers living in mountainous areas because first, mountainous areas are often located in more environmentally vulnerable regions where people, crops, and land are more susceptible to natural changes [4,5]; second, the livelihoods of most smallholders in mountainous areas heavily depend on agriculture, and agricultural losses caused by hazards reduce the ability of the farmers to cope with the risks [6].

Humans must learn to “coexist with hazards” in their relations with nature; adopting appropriate adaptation strategies for geo-hazards is the primary method to mitigate their impact. Initially, the concept of “adaptation” emphasized that the natural selection of the environment allows species to evolve and that humans adjust their behaviors to avert the impact of hazards [7,8]. Subsequently, scholars proposed the concept of “adaptive hazard mitigation”, which is the perspective that human management actions should be viewed as multidimensional experiments with an associated need for post experiment monitoring, evaluation, learning, and adjustment to respond to actual, perceived, or expected environmental changes and their impacts [9]. The behaviors adapted to the geo-hazards addressed in this study are defined as those in which the farmers chose to adjust their livelihood strategy to adapt to the impact of the hazards by comprehensively considering their past experiences and external social influences.

We chose the mountainous areas of Chongqing City as the study area for several reasons. First, Chongqing is a typical mountainous city in China, where is also one of the cities most seriously threatened by mountain disasters. Second, Chongqing has a huge rural population (21.7122 million), accounting for approximately 71% of the total population of Chongqing (Chongqing statistical yearbook, 2017), and most of the rural settlements are located in those disaster-prone areas. Additionally, the lands in Chongqing are not suitable for scale operations due to their fragmentation. Farmers there largely self-decide agricultural choices, thus, encouraging farmers to adjust is a very effective adaptation strategy to geo-hazards. Taken together, in Chongqing, a city with frequent geo-hazards and the most threatened population, it is necessary to investigate the factors affecting farmers’ adjustment to agricultural production behavior. This paper focuses on farmers’ disaster experiences and the role of external social support to comprehensively reflect the impact mechanisms of the farmer’s adaptive behavior to geo-hazards. Through the elucidation of farmers’ choice behaviors and motivations, the results of this study can provide implications for developing more effective geo-hazards risk management programs. The main topics of this study are as follows: What is the main adaptation strategy that farmers use to cope with geo-hazards? What are the main obstacles to disaster risk reduction? What are the factors affecting the choice among different adaptation strategies to geo-hazards?

## 2. Literature Review

### 2.1. Adaptation Strategies to Hazards

Natural hazards have a negative impact on agriculture [10,11,12], and adaptation is considered a choice among behaviors that cope with the negative impact of hazards [13,14]. Since the 1990s, the vulnerability of different populations when facing hazards has been emphasized in the study of disaster adaptation, while the details of individuals’ active adaptation have been given importance [15]. Scholars have found that the adjustment of agricultural production has been a common adaptation strategy [16].

Common agricultural adjustment strategies can be categorized into the following three categories:

The first category is improvement through diversification, including diversification of crop varieties and income sources. The diversification of crop varieties refers to selecting new crop varieties in an environment with potential hazard dangers to cope with natural hazards based on crop improvement and adjustment, e.g., choosing drought-tolerant varieties in arid areas [17]; practicing intercropping, e.g., rice-fish rotations; or planting crops with varying disaster-tolerating levels to reduce the risk of disaster losses and thus provide basic protection for farmers’ livelihoods [10,18]. The diversification of income sources refers to the shift from traditional agricultural livelihoods to off-farm activities [2], e.g., performing labor division among family members or becoming migrant workers to improve livelihood resilience and ensure stability and an increase in smallholder income.

The second category is crop management, referring to the changing of planting dates and locations. The purpose of changing planting dates is to change the length of a crop’s growth period or planting and harvesting dates so the critical stages of crop growth can avoid the peak period for a natural hazard event, e.g., planting crop varieties that mature before the start of the flooding season in a flood-prone region [19,20]. This adaptation strategy is more suitable for natural hazards with certain periodicity and regionality. By spatially separating agricultural products from geo-hazards, farmers can invest their limited production materials in safer areas. The core of crop management is to avoid high-disaster risk areas spatially or avoid high-disaster risk time periods temporally. 

The third category is an adaptation from the perspective of improving productivity, including an increase in vegetation coverage and improvement in soil conservation and irrigation, representing a defensive adaptation strategy. Improving irrigation increases agricultural productivity by replenishing rainwater during a dry season, allowing farmers to mitigate crop losses caused by a drought [19,21].

In short, a farmer’s adaptation enables crops to temporally and spatially avoid hazard threats by adjusting agricultural inputs and employment choices based on the understanding of the space-time patterns of natural hazards. Adaptation strategies can be heterogenous, i.e., different types of hazards, geographical environments, hazard-affected populations, and stages of hazard development have different adaptation strategies. In this study, adaptation to geo-hazards means that to cope with various negative effects of geo-hazards, farmers adjust the distribution of productive resources, such as land and capital, to sustain or improve their current living conditions. Because soil conservation and tree planting methods are not relevant to this study, the proposed adaptation strategies refer only to the first two categories.

### 2.2. Factors that Influence Adaptation strategy

Adaptation is the comprehensive outcome of the combined effects of individual smallholders (Figure 1), the social environment, and government policies and is thus affected by various factors such as the internal factors of the smallholders (e.g., disaster experience) and external factors (e.g., social support).

Disaster experience is an important factor affecting disaster avoidance behavior [22,23]. It was found that individuals’ personal experience with disasters against the backdrop of various changes, such as climate change and floods, prompts individuals to adapt, e.g., a smallholder that has suffered a severe loss because of climate change is more likely to change crop varieties to cope with climate change [18]. The victims of floods realize that they are more susceptible to flooding and are more concerned and apprehensive about flooding; thus, they are more willing to take adaptive action [24]. In addition to disaster experience, disaster severity can largely explain farmers’ adaptation behavior [25,26]. Moreover, some studies showed that the number of disasters experienced has an impact on adaptation [24,27]. 

Many studies revealed that governments, media, and other social networks can enhance the farmers’ ability to adapt to natural hazards and influence their choice of adaptation strategies [28,29]. Obtaining subsidies from the government and governmental credit agencies can alleviate the financial barriers to coping with hazards, and financial support increases the farmers’ likelihood of adopting adaptation strategies [30,31]. Government promotion policies likely encourage farmers to change their farming practices in response to natural hazards [32]. In addition, for the relatively secluded farmers in the mountainous areas, governments and media agencies are their main sources of information, and correct information about hazard and agricultural production can improve the farmers’ ability to cope with natural risks, while poor prediction information is detrimental to farmers [33]. Social networks, often represented by the number of relatives and neighbors that a farmer has in the local area, help individuals adapt to risks [34,35]; such social networks through the links of geography and clan can alleviate the dilemma of hazards adaptation.

Differences in basic characteristics of individuals and smallholders when faced with the same natural environment stimulus lead to different coping behaviors. It was found that men are more likely to obtain information about new technologies and access to resources; thus, they are more likely to change their livelihood strategies than women [36]. However, some scholars argued that women are more sensitive to climate change and thus are more likely to adopt adaptive behaviors [37,38]. People with higher education levels are more inclined to take initiative to cope with natural hazards [36]. It was also found that older farmers are more conservative in their production and are not willing to take the risks associated with new farming techniques [39]. Farmers with lower incomes or high dependence on agricultural income are likely to adopt their original mode of production to ensure their basic survival, while wealthier farmers have more opportunities to access information and loans and have longer-term personal plans [40]. In addition, farmers’ strategies of coping with natural hazards also depend on their family size; they may be forced to transfer part of their labor to off-farm activities [41].

In summary, an individual’s behavioral adaptation to hazards has been extensively investigated. However, first, in terms of hazard type, past studies have been mostly focused on climate change and primarily wide-area natural hazards such as floods and droughts. These studies have improved our understanding of how farmers cope with climate change but have rarely addressed the issue of “farmers’ adaptation under the impact of geo-hazards”. Differing notably in time scale, impact scope, and impact pattern with other natural disasters, geo-hazards are characterized by locality, sporadicity, and grave destructiveness. Currently, adaptation to geo-hazards has been rarely studied from the perspective of individual behavior. Due to different feelings of and negative impacts on farmers caused by different types of natural hazards, farmers’ hazard adaption behaviors may vary. Second, in terms of research areas, past studies have primarily focused on plains and basins and rarely examined mountainous areas, especially the mountainous areas that have a large population in China. Moreover, farmers’ internal driving factors and external incentives for adaptation strategies have not been simultaneously examined. Since adaptation is the comprehensive outcome of a farmer’s own experience and external social influences, both must be included in the adaptation model.

## 3. Study Area and Sample Data

### 3.1. Study Area

The data for this study was gathered from a questionnaire survey conducted from August to October 2018 in geo-hazards prone areas in Chongqing. We selected four sample counties (districts) in Chongqing with a high frequency of geological hazards, namely, Wanzhou District, Yunyang County, Fengdu County and Zhongxian County. In selecting sample townships, we considered the difference in levels of economic development and ultimately chose 18 townships; in each township, one to five villages that have been threatened most seriously by geo-hazards were chosen as samples; and in each village, 15 to 25 smallholders were randomly sampled (see Figure 2). Ten trained interviewers conducted face-to-face interviews as part of the questionnaire survey. Ultimately, 516 valid questionnaires were recovered; the recovery rate was 100%.

### 3.2. Sample Data

In the questionnaire survey, respondents were asked about which adaptation strategies (e.g., no adaptation; adjusting crop varieties, i.e., diversification of crop varieties; reducing farming acreage and diversifying into off-farm employment, i.e., diversification of income source; changing planting dates; and changing planting sites) have been taken when coping with the impact of geo-hazards. It was found that 49.2% of the surveyed smallholders have adopted adaptation strategies to cope with the threat of geo-hazards in the mountainous areas, while 50.8% of the smallholders have not made any immediate adjustments.

As shown in Figure 3, among the smallholders adopting an adaptation strategy, 31.6% of the smallholders chose to change cultivar types, where the farmers changed the crop varieties based on their own experiences. About a quarter of the smallholders chose to find a new job in the non-agricultural industry for securing livelihoods in response to geo-hazards. The adjustment of planting dates was the most difficult adaptation method to achieve and was only adopted by 17.8% of the surveyed smallholders. Influenced by external factors such as soil and climate, the growth period of crops is relatively stable; although it is possible to reduce the loss from flooding by choosing a planting date to avoid the flood-prone rainy season, there are still risks of yield decrease. Compared to China, many farmers who lived in other hazard-prone areas in South-East Asia have different adaptation choices due to disaster types, disaster likelihood and severity, land conditions, policy, etc. For instance, according to a survey conducted in a flood-prone area of the eastern Indian state of West Bengal, Bhattacharjee et al. found that more than 90% of rural households shifted to non-agricultural work in the wake of the increasing flood. This difference stems from a rural employment guarantee program in India [15]. While in Vietnam, scholars revealed that changing crop varieties was one of the most popular practices, and switch to new crop varieties was seldom mentioned [18].

Furthermore, the respondents who had not made any adaptation were asked to describe the main obstacles (e.g., lack of information, lack of funds, lack of labor, lack of land, lack of companions to adaptation, i.e., peer effects) that have prevented them from adjusting their agricultural production; those who had adapted were asked to describe their largest difficulty in adopting adaptation strategies to geo-hazards. The results are shown in Figure 4. Regardless of the presence or absence of adjustment of agricultural production, the largest adaptation obstacle faced by the farmers was a lack of funds, while the lack of companions to adaptation was the lowest factor. Among the smallholders that had adopted an adaptation strategy geo-hazards, 110 (43.5%) claimed that the difficulty was a lack of funds, and only 10 (4.0%) believed they did not adjust merely because others did not. Among the smallholders that had not adopted any adaptation strategy, 92 (35.0%) claimed that the largest obstacle was a lack of funds. Compared with the smallholders that had adapted, those that had not adapted showed higher levels in lack of labor and lack of access to off-farm work and were prone to make their choice based on the choices made by others, i.e., they were more influenced by the peer effect. Moreover, in terms of the factors affecting the adaptation strategy in mountainous areas, the shortage of labor was second to the shortage of funds. Because of an increase in urbanization, many rural laborers have migrated to urban areas, which, coupled with the threat of geo-hazards, has forced some farmers to abandon farming. These findings are widely consistent with other researches adopted in neighboring countries. For example, Alauddin et al. and Alam et al. recognized that limited access to information about potential climate change and drought-resistant rice varieties, limited access to credit or funds, limited or lack of land ownership, etc. were major barriers to adapt in hazard-prone areas of Bangladesh [19,42]. However, the barriers varied in different countries. In Bangladesh, lack of information was the major barrier, while the labor shortage was relatively less important.

## 4. Empirical Model and Explanatory Variables

### 4.1. Econometric Model

In this study, we attempted to analyze farmers’ adaptation strategies in mountainous areas threatened by geo-hazards and focused on their adaptation strategies in terms of agricultural production. To better quantify the effects of different factors on farmers’ adaptation strategies, we constructed two models according to the characteristics of the dependent variable, a binary logit model (Model 1), and a multinomial logit model (MNL, Model 2).

The binary logit model (Model 1) has been widely adopted because it has analytical advantages in dealing with discrete binary outcomes. This study used a binary logit model to analyze various factors affecting farmers’ decisions to apply adaptation strategies to geo-hazards in agricultural production. A farmer’s decision to apply adaptation strategies is of a discrete choice form. The general form of a binary logit model is as follows [43]:(1)$$\mathrm{Pr}(y_{i}=j)=\frac{e^{X\beta }}{1+e^{X\beta }}$$
where $\mathrm{Pr}\left(y_{i}=j\right)$ indicates the probability of farmer *i* adopting adaptation strategy *j*; *j* is the farmers’ adaptation strategy, *j* = {0, 1} is a set in which 0 denotes farmers who did not adapt to a geo-hazard and 1 denotes farmers who adapted to a geo-hazard in their agricultural production; *β* is the vector of parameters; and *X* is the vector of the affecting factors. 

The distinguishing feature of Model 2 is it analyzes the discrete selection problem of respondents in a set of different hazard adaptation schemes. Because the valuation of the dependent variable is polynomial, disordered, and discrete, the multinomial logit model has been used to analyze this selection problem and has also been widely used in the study of individuals’ choice behavior [44,45]. The adaptation strategies for farmers to choose were categorized into the following five categories: no adaptation; adjusting crop varieties; reducing farming acreage and diversifying into off-farm employment; changing planting dates; and changing planting sites. Thus, the probability that Farmer *i* chooses choice *j* is as follows:(2)$$\mathrm{Pr}(y_{i}=j)=\left\{\begin{array}{l} \frac{1}{1+\sum_{k=1}^{4}\exp(x_{i}^{\prime }\beta_{k})},(j=0)\\ \frac{\exp(x_{i}^{\prime }\beta_{j})}{1+\sum_{k=0}^{4}\exp(x_{i}^{\prime }\beta_{k})},(j=1,\ldots ,4) \end{array}\right\}$$

The meanings of $\mathrm{Pr}\left(y_{i}=j\right)$, *β* and *X* are defined as binary logit model above. While in multinomial logit model, *j* = {0, 1, 2, 3, 4} is the farmers’ adaptation strategy set, in which 0 = no adaptation, 1 = adjusting crop varieties, 2 = reducing farming acreage and diversifying into off-farm employment, 3 = changing planting dates, and 4 = changing planting sites, the choice of Farmer *i* must be in the adaptation strategy set. In this study, the choice of 0 (no adaptation) was used as the reference group and its coefficient was set to $\beta_{0\;}$= 0.

In Model 2, unbiased and consistent parameter estimates of the MNL model require the assumption of the independence of irrelevant alternatives (IIA) property, which states that the probability ratio of any two alternatives is independent of the attributes of any other alternative in the choice set [46].

In both models, the parameters of the model cannot be directly interpreted. In particular, a positive coefficient does not necessarily mean that an increase in the value of the coefficient of an explanatory variable will cause an increased probability of the choice. Marginal effects measure the likely change in the probability of the adaptation of a particular choice with respect to a unit change in an explanatory variable. The marginal effects are usually derived as follows [43]: (3)$$ME_{ijk}=\frac{\partial \mathrm{Pr}(y_{i}=j)}{\partial x_{ik}}$$
where $ME_{ijk}$ refers to the effects of the kth explanatory variable on the probabilities (farmer *i* chooses adaptation strategy *j*).

### 4.2. Selection of the Explanatory Variables

The explanatory variables were selected based on the literature analysis, theoretical analysis, and data availability [3,15,18,29,34,38,39]. The explanatory variables used in this study included personal and smallholder characteristics, social support factors, and disaster experience factors. The personal and smallholder characteristics include gender, smallholder family size, smallholder age, smallholder education level, farming acreage, agricultural income, and main livelihood of the smallholder. An in-depth description of the explanatory variables is presented in Table 1.

#### 4.2.1. Social Support Factors

We regarded the individuals and formal and informal organizations that can provide material or nonmaterial resources for farmers as the farmers’ social support system, mainly referring to the funding, information, and project assistance to farmers by the government, the media, and relatives for adjusting agricultural production. Specific variables included state subsidy, maximum loan amount, main information source, government information effectiveness, presence of disaster prevention construction, the role of media information, the number of relatives that will lend money, and the number of relatives who can introduce off-farm employment.

Specifically, state subsidy refers to the amount of money that the government transfers from the national Treasury to farmers, which works as a direct supplement to farmers’ cash flow. The maximum loan amount refers to a ceiling on how much a family can borrow from a bank, and an increase in the maximum loan amount relaxes the cash constraint on smallholders, enabling farmers to adjust crop varieties and improve facilities. These two variables reflect the financial support of farmers from formal institutions. The main information source is measured by the question that “what is the main information channel?” with dummy options (1 = farmers’ main information source is the government, 0 = otherwise). The government information effectiveness is evaluated as “whether local governments can provide effective information for farmers after a disaster?”. The variables about information are purported to examine the existence and effectiveness of the government’s information transfer function. The presence of disaster prevention construction refers to whether the geo-hazard site has any prevention construction, which reflects the government’s efforts in disaster risk management. Because disaster prevention and control construction can mitigate the possibility and calamity of disasters, they can indirectly affect farmers’ agricultural production behavior. The role of the media information is reflected through “do you often watch television, read the newspaper, or browse the Internet for news on hazards?”. Both traditional and new media are an important way for farmers to obtain information, such as disaster forecasting and agricultural adjustment information. The number of relatives that will lend money and the number of relatives who can introduce off-farm employment is represented by the number of relatives who can borrow money or help the family find a job in non- agricultural industry. These two measurements reflect the informal support a farmer can obtain for agricultural adaptation. All those social support factors have the potential to facilitate agricultural adaptation when concerning geo-hazards.

#### 4.2.2. Disaster experience factors

To examine the impact of an individual’s disaster experience on the choice of agricultural adaptation strategies to geo-hazards, we chose the following variables: the number of disasters experienced, disaster-caused property damage experience, and crop loss experience.

An important indicator for measuring disaster experience is the number of geo-disasters an individual or a smallholder has experienced. The disaster-caused property damage experience and crop loss experience are measured by whether the family has suffered any financial or physical loss (direct and indirect economic losses) from geo-hazards event, respectively. This study supposes that more disaster experiences and more losses result in the greater the psychological impact; thus, the individual or the smallholder would attach more importance to a geo-hazard when making agricultural decisions. The description of the explanatory variables is shown in Table 1.

## 5. Modeling Results and Discussion

### 5.1. Estimation of Parameters

In the parameter estimations of the two models in this study, the adaptation strategy = 0 (not adjusting) was used as the reference group. Table 2 shows the estimated coefficient values and *p*-value.

Table 2 shows the coefficient estimates for the two models, as well as the directions of the impacts of personal and smallholder characteristic variables, social support variables, and disaster experience variables on adaption strategy choice. The estimation results of Models 1 and 2 showed that, first, the likelihood ratio statistics of the two models were all significant at the significance level of 1% and have passed the chi-square test (Table 2). The Pseudo R2 values of Models 1 and 2 were 0.1776 and 0.1772, respectively, indicating that the models can be used in a multivariate analysis of the cross-sectional data. The variance inflation factor (VIF) values of all the variables were lower than 10 (1.03~1.30), indicating the absence of serious multicollinearity. Second, we tested the IIA by employing the Hausman test. The test result failed to reject the null hypothesis of IIA at the 5% level (x2 ranged from −12.03 to 60.56, with probability values ranging from 0.85 to 1.00), indicating that the models have strong explanatory power.

Farmers’ personal or smallholder characteristics may affect their decisions on adaptation strategy. The coefficients in Model 1 describe the relationship between the explanatory variable and whether an adaptation strategy is adopted, while those in Model 2 further illustrate the relationship between the explanatory variable and which adaptation strategy is adopted. For example, that the sign of agricultural income in Model 1 is positive and statistically significant means that when other factors are kept constant, the higher the agricultural income, the more likely the farmer will adopt an adaptation strategy. Model 2 shows that the higher the agricultural income, the more likely the farmer will adopt the four adaptation strategies. Moreover, the signs of the coefficients of family size, smallholder age, smallholder education level, farming acreage, and smallholder livelihood were all positive in Model 1, indicating that the larger the smallholder family size, the older the smallholder, the higher the education level of the smallholder, and the larger the farming acreage, the more the smallholder livelihood relies on agriculture and the more likely the smallholder will adopt an adaptation strategy. Model 2 shows that smallholders with a large family size were more likely to reduce farming acreage to engage in other employment or choose to change planting dates to avoid the peak disaster times to reduce disaster-caused loss. However, smallholders with a large family size are generally reluctant to adjust crop varieties, likely because the inertia from their long production experience makes them more willing to remain with previous crop varieties. Moreover, the age of the smallholder had a positive impact on the adjusting of crop varieties, planting dates, and farming acreage. However, the older the smallholder, the lower the probability of reducing the planting area and engaging in off-farm employment. This is because older farmers lack the means of shifting to other jobs and are more reliant on agricultural production at both the psychological and practical levels. The influence direction of basic characteristics on natural hazards adaptation was basically consistent with the findings of other studies [15,32].

Second, the regression analysis results showed that social support and disaster experience can significantly encourage farmers to adopt an adaptation strategy. The results of Model 1 showed that social support could significantly encourage farmers to adjust agricultural production; financial support exerts a very significant impact on adaptation strategy choice, which is consistent with the farmers’ claim that “the greatest adaptation obstacle is a lack of funds”. Whether a village has disaster prevention construction could significantly increase the enthusiasm of the villagers in adopting adaptation strategies. The sign of the coefficients of the government’s information support and media information role indicated that information communication could increase farmers’ likelihood of adopting an adaptation strategy. Most of the variables of disaster experience were highly significant, and after controlling the influence of other variables, the disaster-caused smallholder property and crop losses can significantly increase the likelihood of farmers to adjust agricultural production.

The results of Model 2 further indicated that state subsidies and disaster prevention construction encouraged the adoption of the four types of adaptation strategy. The funding and project support from the government reflects that the government has attached importance to disaster risk management while providing direct assistance to farmers in adaptation strategies. Government information services reduced the likelihood of choosing the adaptation strategy of “reducing farming acreage and engaging more in off-farm employment”; however, the information from the media had the opposite effect, likely because the government has been more involved in disseminating information about disaster forecasting, prevention, and agricultural management, while the use of smartphones, the Internet, and other media means that farmers are exposed to a wider outside world. It is more convenient for smallholders that have accepted information from the media to change from farming to other professions. Support from relatives enhanced the farmers’ adaptability, exerting a positive effect on all four adaptation strategies, and the relatives that can lend money and introduce a job provided the farmers with the funds and information needed for agricultural production adaptation. Moreover, due to the instability of geo-hazards and the uncertainty of disaster-coping outcomes, many smallholders were inclined to seek means to increase non-agricultural income to improve their adaptive capacity.

Similarly, a disaster experience had an important impact on which adaptation strategy a smallholder would choose. The signs of the variable coefficients of Model 2 indicate that the number of disasters experienced with disaster-caused property damage and crop loss generally increased the likelihood of choosing the four types of adaptation strategy, suggesting that disaster losses that have been suffered by individuals or smallholders enhance their adaptation to geo-hazards.

### 5.2. Marginal Effects Results

The parameter estimations of Models 1 and 2 only provided the direction of the impact of the independent variables on the dependent variable; the values of the parameters do not necessarily reflect the actual levels of the impacts. The marginal effects from the binary logit model and MNL model measured the expected change in probability of a particular choice being made with respect to a unit change in an independent variable. In all cases, the estimated coefficients should be compared with the base category of no adaptation [32]. Therefore, we examined the marginal effects of the binary logit and the MNL models, which, along with their respective *p* values, are shown in Table 3.

In Table 3, the coefficients of the marginal effects of Model 1 indicate that the smallholder family size increased the likelihood that the smallholder adopts an adaptation strategy by 1.0%. Specifically, for each one unit increase in the smallholder family size, the likelihood of adjusting crop varieties was decreased by 0.5%, of reducing farming acreage was increased by 1.2%, of adjusting planting dates was increased by 0.8%, and of changing planting sites was increased by 0.2%. Among personal and smallholder basic characteristics, agricultural income was the factor that has the most significant impact on the smallholder’s adoption of an adaptation strategy. For each 10,000 yuan increase in agricultural income, the likelihood of adopting an adaptation strategy was increased by 1.2%, and especially, the flexibilities of choosing to adjust crop varieties (1.3%) and planting sites (1.8%) were significantly greater than those of choosing the other two strategies. In addition, in Model 2, the marginal effect coefficients of the gender, age, and education level of the smallholder, and farming acreage with adaptation strategy 2 were −0.016, −0.001, −0.007, and −0.006, respectively. Reducing farming acreage to engage more in off-farm jobs is, to some extent, similar to abandoning agriculture. In the cases where the smallholder is male, older, and owns a large amount of farming acreage, the smallholder was more heavily reliant on agriculture.

In addition, Table 3 shows that government financial support had a highly significant impact on the adoption of an adaptation strategy. This result is consistent with many studies performed in Asia, which suggested that monetary resource availability avails smallholders to improve their financial situation and thus they can meet transaction costs for adaptation [18,47,48]. Specifically, for each 10,000 yuan increase in state subsidies and maximum loan amount, the likelihood of adopting an adaptation strategy was increased by 3.0% and 0.2%, respectively, and the impact of state subsidies was greater than that of the maximum loan amount, indicating that the effect of the direct financial support of the government is more profound than that of an indirect loan. The government information support and project support could increase the likelihood of adopting an adaptation strategy by 4.0% and 2.4%, respectively. If the main source of information for farmers is the government, the likelihood of reducing farming acreage and changing to off-farm jobs was decreased by 4.6%, indicating that information from the government can encourage farmers to make agricultural production adjustments to adapt to geo-hazards. The role of the information from the media was able to increase the likelihood of adopting an adaptation strategy by 4.3% and that of adjusting an agricultural area by 7.2%. The positive influence of information provided by the government or media is also highlighted by Trinh et al., who identified that information about potential environmental risks and adjustment of agricultural production would promote adaptive strategy adopted by farmers in Vietnam [18]. Financial and information support from relatives facilitated farmers’ agricultural production adjustments; for each one person increase in the number of relatives that will lend money, the likelihood of adjusting crop varieties was increased by 2.3%, since the accessibility to financing reduced the difficulty in purchasing materials for agricultural production. As the number of relatives capable of introducing a job increases, the likelihood of choosing adaptation strategy 2 increased by 0.7%. In rural areas, those who can introduce people to various jobs are generally governmental employees who can prompt farmers to better connect with outside communities, reducing the cost of changing to non-agricultural livelihoods.

It is also noteworthy that farmers’ disaster experience plays an important role in their choice of adaptation strategies. The estimation results based on Models 1 and 2 indicate that the significance level of the number of disasters experienced was not high, with a low elastic coefficient (0.2%), likely because the number of disasters experienced does not mean that the disasters have had any actual impact on the farmers and thus exert little effect on their agricultural adaptation strategy. This is consistent with the results of other studies. However, if farmers have experienced disaster-caused property or crop loss, their willingness and flexibility to adopt adaptation strategies would be significantly increased. Similar findings have been given by Acosta et al. and Trinh et al., in that people affected by previously floods and landslides suffered psychological damage, which may lead them to adaptation options in the Philippines and Vietnam [18,49]. We also found that the experiences of disaster-caused property and crop losses increased farmers’ likelihood of adopting adaptation strategies by 5.1% and 3.7%, respectively. The disaster-caused property loss increased the farmers’ likelihood of reducing farming acreage and changing to other occupations by 12.2%, which was higher than the impact of the disaster-caused crop loss on this strategy (6.4%); this is because farmers are more sensitive to direct than indirect economic loss.

## 6. Conclusions and Policy Implications

In this study, based on the data collected from the questionnaire survey of rural smallholders in Chongqing City, China, in 2018, we examined the disaster mitigation strategies and barriers of the smallholders in mountainous areas threatened by geological hazards. We found that in these areas, 49.2% of the respondents have adopted agricultural production adjustments to adapt to the threat of geo-hazards. The most common strategy was adjusting crop varieties, followed by reducing farming acreage and changing to other professions, changing planting sites, and adjusting planting dates. Additionally, the farmers claimed that the largest adaptation obstacle was a lack of funds.

We adopted the binary logit and MNL models to investigate the factors affecting farmers’ choice of adaptation methods for geo-hazards. The marginal analysis results showed that smallholder characteristics, social support, and disaster experience could significantly increase farmers’ adaptation to geo-hazards. In smallholder characteristics, agricultural income was the most significant factor affecting farmers’ choice of adaptation strategy. The government’s provision of financial support, information support, and direct disaster prevention construction to farmers have contributed to the farmers’ adoption of the four adaptation strategies. Media information could increase the possibility of farmers’ adjusting agricultural production, especially changing planting sites, to adapt to geo-hazards. Support from relatives could also significantly facilitate farmers to adapt. In addition, farmers’ disaster experience had a significant impact on their choice of adaptation, while the number of disasters experienced generally had little effect on their choice. Farmers’ responses to direct and indirect losses were different, and they were more sensitive to property loss than to crop loss, but both property and crop losses caused by geo-hazards increased the possibility of farmers’ choosing various adaptation strategies to geo-hazards.

The results of this study have certain policy implications and can provide a reference for formulating disaster reduction countermeasures for the mountainous areas plagued by geo-hazards. First, we should increase farmers’ income and enhance their adaptability to geo-hazards. By increasing agricultural subsidies and promoting the implementation of agricultural insurance and loans, the farmers are encouraged to actively cope with geo-hazards. Second, we should strengthen the social support system for farmers and transform the social support into tangible livelihood capital. The government must establish an effective disaster warning mechanism and provide practical guidance to farmers for adopting different adaptation strategies, and release information about the disaster and agricultural production through radio, television, and mobile phone to guide farmers to rationally plan their production. We should also attach importance to the establishment of cooperation mechanisms among farmers and increase off-farm job opportunities for farmers. Third, we should rationally guide the impact of disaster experience on farmers and improve their awareness of geo-hazards to enhance their confidence in adopting adaptation strategies to geo-hazards.

## Figures and Tables

**Figure 1 ijerph-17-01226-f001:**
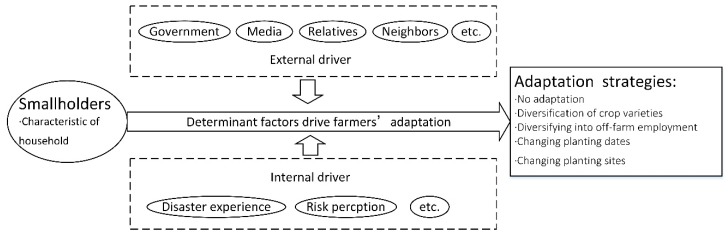
Diagram of the influence mechanisms of farmers’ adaptation.

**Figure 2 ijerph-17-01226-f002:**
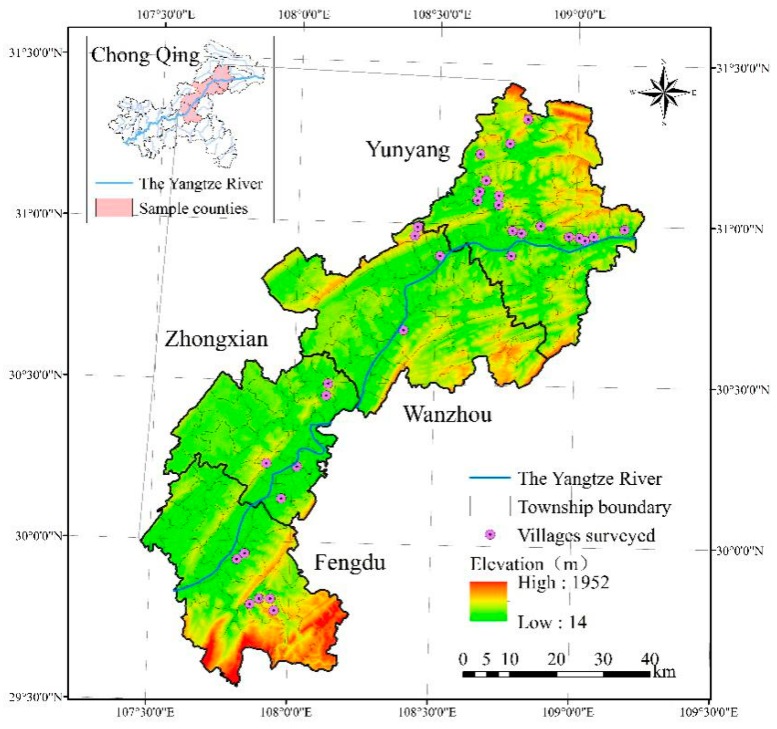
Locations of the sample villages.

**Figure 3 ijerph-17-01226-f003:**
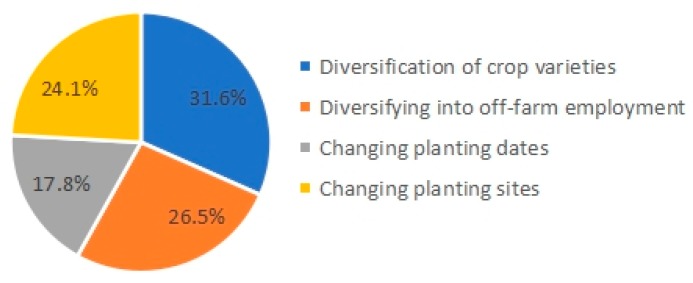
Respondents’ main adaption behaviors.

**Figure 4 ijerph-17-01226-f004:**
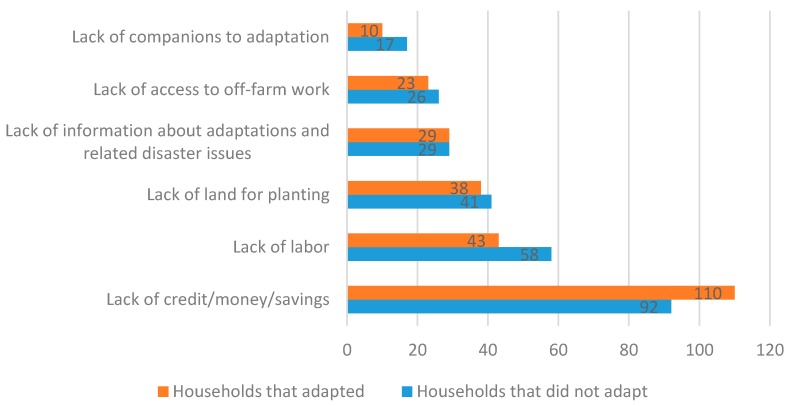
Barriers to adaptation strategies (% of respondents).

**Table 1 ijerph-17-01226-t001:** Description of explanatory variables that affect adaptation strategies.

Variable	Description	Mean	S.D.
Gender	The gender of the respondent. Dummy (male = 1, female = 0)	0.448	0.498
Family size	The number of your family members. Discrete (number)	4.155	1.943
Smallholder age	The age of the smallholder. Discrete (years)	59.432	11.265
Smallholder education level	The schooling years of the smallholder. Discrete (years)	5.434	3.741
Farming acreage	The acreage of cultivated land. Continuous (mu)	6.185	24.314
Main livelihood of the smallholder	What is your mainly major source of income? Dummy, (Agriculture = 1; non- Agriculture = 0)	0.790	0.407
Agricultural income	The family annual agricultural income. Continuous (10,000 yuan)	1.874	1.727
State subsidy	The amount of state subsidy your family receive a year. Continuous (10,000 yuan)	2.109	1.494
Maximum loan amount	How much money can your family borrow from the bank? Continuous (10,000 yuan)	2.421	2.366
Main information source	What is the main information channel? Dummy (Government = 1; otherwise = 0)	0.558	0.497
Effectiveness of government information	Whether local governments can provide effective information for farmers after a disaster? Dummy (yes = 1; no = 0)	0.587	0.975
Presence of disaster prevention construction	Is there a disaster prevention construction? Dummy (yes = 1; no = 0)	0.602	0.572
Role of media information	Do you often watch television, read the newspaper, or browse the Internet for news on hazards? Dummy (yes = 1; no = 0)	0.624	0.485
Number of relatives who will lend money	The number of relatives who would lend money if necessary. Discrete (number)	2.645	1.179
Number of relatives who can introduce off-farm employment	The number of relatives who can introduce off-farm employment if necessary. Discrete (number)	2.187	2.697
Number of disasters experienced	The number of disasters experienced in the current lifetime. Discrete (number)	6.465	9.849
Property damage experience	Is there any disaster-caused property damage? Dummy (yes = 1; no = 0)	0.787	1.410
Crop loss experience	Is there any disaster-caused crop loss? Dummy (yes = 1; no = 0)	0.378	0.485

S.D.: standard deviation; mu: a unit of area, 1 mu equals 0.4 hectares.

**Table 2 ijerph-17-01226-t002:** Parameters estimation results on the determinants of adaptation strategies.

Variables	Model 1: Binary Logit		Model 2: Multinomial Logit (Reference Group = not Adjusting/no Adaptation)							
	Adaptation Strategy (1 = yes)		1: Adjusting Crop Varieties		2: Reducing Farming Acreage		3: Changing Planting Dates		4: Changing Planting Sites	
	Coef.	*p*	Coef.	*p*	Coef.	*p*	Coef.	*p*	Coef.	*p*
Gender	0.310	0.163	0.551 *	0.083	−0.056	0.873	0.628	0.255	0.284	0.442
Family size	0.082	0.147	−0.004	0.965	0.171 *	0.051	0.272 **	0.037	0.060	0.522
Smallholder age	0.017 *	0.098	0.035 **	0.027	−0.008	0.623	0.012	0.636	0.015	0.420
Smallholder education level	0.023	0.461	0.079 *	0.066	−0.072	0.143	0.070	0.340	−0.007	0.890
Farming acreage	0.007 *	0.089	0.011 **	0.013	−0.069	0.130	−0.007	0.749	0.009	0.157
Main livelihood of the smallholder	0.042	0.881	-0.345	0.368	0.937 *	0.071	−0.414	0.521	0.056	0.901
Agricultural income	0.188 ***	0.005	0.231 **	0.022	0.190 *	0.071	0.115	0.500	0.329 ***	0.007
State subsidy	0.339 **	0.044	0.322	0.102	0.000	0.221	0.439	0.105	0.418 *	0.052
Maximum loan amount	0.023 **	0.012	0.036 ***	0.002	0.287 ***	0.005	−0.063	0.253	−0.009	0.688
Main information source	0.303	0.175	0.838 **	0.011	−0.035	0.276	0.646	0.240	0.157	0.676
Effectiveness of government information	0.179	0.113	0.333 **	0.045	−0.146	0.403	0.164	0.557	0.334 *	0.078
Presence of disaster prevention construction	0.333 **	0.014	0.419 **	0.044	0.210	0.303	0.283	0.409	0.444 *	0.063
Role of media information	0.372	0.126	0.416	0.245	0.072	0.850	−0.209	0.734	1.084 ***	0.008
Number of relatives that will lend money	0.179 *	0.075	0.293 **	0.045	0.075	0.648	0.245	0.305	0.158	0.353
Number of relatives who can introduce off-farm employment	0.118 ***	0.007	0.096 *	0.088	0.131 **	0.027	0.033	0.748	0.163 ***	0.004
Number of disasters experienced	0.012	0.246	0.032 **	0.017	−0.011	0.551	0.005	0.855	0.013	0.421
Property damage experience	0.669 **	0.022	0.266	0.505	1.656 **	0.012	0.375	0.593	0.661	0.195
Crop loss experience	0.999 ***	0.000	0.762 **	0.020	1.123 ***	0.001	0.943 *	0.093	1.261 ***	0.001
Constant	−6.321 ***	0.000	−9.037 ***	0.000	−5.924 ***	0.001	−8.352 ***	0.002	−8.403 ***	0.000
LR chi2	118.51		201.06							
Pseudo R2	0.1776		0.1772							

*, **, and *** indicate significance at 1%, 5%, and 10% probability levels, respectively. LR chi2 refers to Likelihood Ratio Chi-Square.

**Table 3 ijerph-17-01226-t003:** Marginal effects on the determinants of adaptation strategies.

Variables	Model 1: Binary Logit		Model 2: Multinomial Logit (Reference Group = not Adjusting/no Adaptation)							
	Adaptation Strategy (1 = yes)		1: Adjusting Crop Varieties		2: Reducing Farming Acreage		3: Changing Planting Dates		4: Changing Planting Sites	
	dy/dx	*p*	dy/dx	*p*	dy/dx	*p*	dy/dx	*p*	dy/dx	*p*
Gender	0.039	0.160	0.045	0.117	−0.016	0.549	0.016	0.352	0.010	0.691
Family size	0.010	0.145	−0.005	0.529	0.012 *	0.071	0.008 *	0.067	0.002	0.812
Smallholder age	0.002 *	0.095	0.003 **	0.030	−0.001	0.305	0.000	0.821	0.001	0.663
Smallholder education level	0.006	0.460	0.008 **	0.033	-0.007 *	0.070	0.002	0.362	-0.001	0.729
Farming acreage	0.001 *	0.085	0.002 ***	0.008	−0.006	0.115	−0.000	0.901	0.001 *	0.058
Main livelihood of the smallholder	0.050	0.881	−0.042	0.212	0.080 **	0.049	−0.014	0.463	0.001	0.980
Agricultural income	0.012 ***	0.004	0.013	0.136	0.009	0.295	0.001 ***	0.008	0.018 **	0.039
State subsidy	0.030 **	0.042	0.018	0.230	0.013	0.431	0.009	0.216	0.020	0.107
Maximum loan amount	0.002 **	0.010	0.003 ***	0.000	0.003 ***	0.002	-0.002	0.205	-0.001	0.375
Main information source	0.040	0.172	0.077 ***	0.009	−0.046	0.104	0.016	0.325	0.000	0.997
Effectiveness of government information	0.020	0.110	0.027 *	0.071	−0.020	0.138	0.003	0.752	0.020	0.128
Presence of disaster prevention construction	0.024 **	0.012	0.028	0.132	0.006	0.705	0.004	0.675	0.022	0.184
Role of media information	0.043	0.123	0.022	0.491	0.010	0.721	−0.013	0.493	0.072 **	0.012
Number of relatives that will lend money	0.018 *	0.072	0.023 *	0.084	0.000	0.994	0.005	0.468	0.005	0.668
Number of relatives who can introduce off-farm employment	0.008 ***	0.006	0.005	0.313	0.007 *	0.079	0.001	0.872	0.009 **	0.013
Number of disasters experienced	0.002	0.244	0.003 **	0.012	−0.001	0.306	0.000	0.992	0.001	0.631
Property damage experience	0.051 **	0.020	0.008	0.832	0.122 **	0.021	0.002	0.907	0.024	0.502
Crop loss experience	0.037 ***	0.000	0.033 *	0.063	0.064 **	0.013	0.017 **	0.012	0.064 **	0.013

*, **, and *** indicate significant at 1%, 5%, and 10% probability level, respectively.

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
