# Peer review of "Understanding the Resilience of Different Farming Strategies in Coping with Geo-Hazards: A Case Study in Chongqing, China"

_ijerph, 2020, doi:10.3390/ijerph17041226_

Round 1
Reviewer 1 Report
1. The selected groups of explanatory variables should be explained in more detail (4.2.1 and 4.2.2).
2. In the opinion of the reviewer, it is desirable to give the symbols of variables included in mathematical dependencies (1)-(3) directly after the formulas.
3. It is interesting to compare the obtained for China results with neighboring countries in South-East Asia.
Author Response
Response to Reviewer 1 Comments
Point 1: The selected groups of explanatory variables should be explained in more detail (4.2.1 and 4.2.2).
Response 1: Thank you. We have added more details about explanatory variables in section 4.2.1, 4.2.2, and Table 1.
Point 2: In the opinion of the reviewer, it is desirable to give the symbols of variables included in mathematical dependencies (1)-(3) directly after the formulas.
Response 2: Thanks for your advice. We have put the meaning of each symbol directly after the formulas.
Point 3: It is interesting to compare the obtained for China results with neighboring countries in South-East Asia.
Response 3: Thank you very much. Your comments are very helpful and have improved the paper considerably. We have supplemented some comparisons to similar studies in other countries in South-East Asia, mainly in section 3.2 and 5.2.

Reviewer 2 Report
Farming strategies is one of the adaptive behaviors to cope with hazard risks. This study focused on the resilience of farmers’ behavioral mechanisms in Chongqing, China to address geo-hazards. The results show that the most common adaptation strategy was adjusting crop varieties, and the largest adaptation obstacle was the lack of funds. Additionally, the age of the smallholder, farming acreage, agricultural income, social support, and disaster experience significantly increased the possibility of farmers adjusting their agricultural production. The most important factors affecting a farmer’s adaptation strategy were smallholder agricultural income, state disaster subsidies, the presence or absence of disaster prevention engineering, the smallholder’s property, and the presence or absence of disaster-caused crop loss.
This study provides implications for the government to formulate disaster mitigation measures and for farming strategies at the smallholder level. It is a good job. But I have some comments.
[1] I suggest to add study area (Chongqing, China) in Abstract.
[2] I suggest the Keywords including geo-hazards, framing strategies, adaptability, Chongqing, China.
[3] Line 31-39 showed China’s geo-hazards and its serious impacts. I don’t think it is necessary here because this study focused in Chongqing, China. It is better to show some number of geo-hazards impacts in Chongqing.
[4] Line 184-192 is the introduction of Chongqing City including total area, mountain area, rural population, per capita GDP and potential geo-hazards sites. As mentioned above, I think these information of Chongqing should be included in Introduction.
[5] Line 58-76, this paragraph looks like literature review. I suggest to include it in part two Literature review.
[6] Line 77-80, “We choose the mountainous areas of Chongqing City (typical mountainous areas in China) as the study area…”, I think the reason of choosing Chongqing should be mentioned here, as my suggestions in [4].
[7] Line 82-85 “The main topics of this study…” and line 176-181 “in this study, we chose the mountainous areas of Chongqing that are populous and have been frequented by geo-hazards as the research area. We examine …” seems to show the aims and tasks of this study. I suggest to combine them in Introduction.
Author Response
Response to Reviewer 2 Comments
Point 1: I suggest to add study area (Chongqing, China) in Abstract.
Response 1: Thank you. We have added the study area in Abstract.
Point 2: I suggest the Keywords including geo-hazards, framing strategies, adaptability, Chongqing, China.
Response 2: Thank you. We have made the changes as recommended.
Point 3: Line 31-39 showed China’s geo-hazards and its serious impacts. I don’t think it is necessary here because this study focused in Chongqing, China. It is better to show some number of geo-hazards impacts in Chongqing.
Response 3: We have deleted part of the description about China’s geo-hazards impacts, and added some disaster data in Chongqing. Because Chongqing is located in southwest China, and the overall situation can also indirectly reflect the situation of Chongqing indirectly, so we kept several numbers of likelihood and impacts of southwest regions and China.
Point 4: Line 184-192 is the introduction of Chongqing City including total area, mountain area, rural population, per capita GDP and potential geo-hazards sites. As mentioned above, I think these information of Chongqing should be included in Introduction.
Response 4: Thank you very much. We have included the information of Chongqing City (total area, mountain area, rural population, etc.) in Introduction.
Point 5: Line 58-76, this paragraph looks like literature review. I suggest to include it in part two Literature review.
Response 5: Thank you. We have deleted this paragraph from the introduction, and considering that the content overlaps with the original literature review, we have incorporated the un-overlapped content into the revised literature review (in the last paragraph in section 2.2).
Point 6: Line 77-80, “We choose the mountainous areas of Chongqing City (typical mountainous areas in China) as the study area…”, I think the reason of choosing Chongqing should be mentioned here, as my suggestions in [4].
Response 6: We have made a supplement for why we chose Chongqing as the study area in line 64-73.
Point 7: Line 82-85 “The main topics of this study…” and line 176-181 “in this study, we chose the mountainous areas of Chongqing that are populous and have been frequented by geo-hazards as the research area. We examine …” seems to show the aims and tasks of this study. I suggest to combine them in Introduction.
Response 7: In light of your comments, we have shifted the sentences in line 176-181 (original version), and combined them with line 82-85 (original version). This revision is shown in the last paragraph in Introduction (revised version). Thank you for helping us think this through and better express our aim.
